# Estimating the economic burden of typhoid in children and adults in Blantyre, Malawi: A costing cohort study

**Fumbani Limani**[1], **Christopher Smith**[1], **Richard Wachepa**[1], **Hlulose Chafuwa**[1], **James Meiring**[1], **Patrick Noah**[2], **Pratiksha Patel**[1], **Priyanka D. Patel**[1], **Frédéric Debellut**[3], **Clint Pecenka**[3], **Melita A. Gordon**[1,4], **Naor Bar-Zeev**[1,5]*

1 Malawi-Liverpool-Wellcome Trust Clinical Research Programme, Blantyre, Malawi, 2 Department of Surgery, Kamuzu University of Health Sciences, Queen Elizabeth Central Hospital, Blantyre, Malawi, 3 PATH, Seattle WA, United States of America, 4 Institute of Infection, Veterinary and Ecological Sciences, University of Liverpool, Liverpool, United Kingdom, 5 Department of International Health, International Vaccine Access Center, Bloomberg School of Public Health, Johns Hopkins University, Baltimore MD, United States of America

* nbarzee1@jhu.edu

**Data Availability Statement:** The dataset is available by contacting the data archivist of the Malawi-Liverpool-Wellcome Trust Clinical Research Programme (MLW). The data reside on

## Abstract

### Background

Typhoid causes preventable death and disease. The World Health Organization recommends Typhoid Conjugate Vaccine for endemic countries, but introduction decisions depend on cost-effectiveness. We estimated household and healthcare economic burdens of typhoid in Blantyre, Malawi.

### Methods

In a prospective cohort of culture-confirmed typhoid cases at two primary- and a referral-level health facility, we collected direct medical, non-medical costs (2020 U.S. dollars) to healthcare provider, plus indirect costs to households.

### Results

From July 2019-March 2020, of 109 cases, 63 (58%) were <15 years old, 44 (40%) were inpatients. Mean hospitalization length was 7.7 days (SD 4.1). For inpatients, mean total household and provider costs were $93.85 (95%CI: 68.87–118.84) and $296.52 (95%CI: 225.79–367.25), respectively. For outpatients, these costs were $19.05 (95%CI: 4.38–33.71) and $39.65 (95%CI: 33.93–45.39), respectively. Household costs were due mainly to direct non-medical and indirect costs, medical care was free. Catastrophic illness cost, defined as cost >40% of non-food monthly household expenditure, occurred in 48 (44%) households.

### Conclusions

Typhoid can be economically catastrophic for families, despite accessible free medical care. Typhoid is costly for government healthcare provision. These data make an economic case

the MLW Data Portal here: https://data.mlw.mw/portal/, and are accessible with appropriate ethical approval and in compliance with Data Sharing Policy of the Malawi-Liverpool-Wellcome Trust Clinical Research Programme.

**Funding:** This work was supported by a grant from PATH, as part of the Typhoid Vaccine Acceleration Consortium (TyVAC). TyVAC is a partnership between the Center for Vaccine Development and Global Health at the University of Maryland School of Medicine, the Oxford Vaccine Group at the University of Oxford, and PATH, an international non-profit. TyVAC is funded by the Bill & Melinda Gates Foundation (OPP1151153). There was no additional external funding received for this study.

**Competing interests:** NBZ is co-investigator on research grants to Johns Hopkins University from Serum Institute of India and from Johnson & Johnson, both outside this work. All other authors declare that they have no competing interests.

for TCV introduction in Malawi and the region and will be used to derive vaccine cost-effectiveness.

## Introduction

Typhoid is caused by a Gram-negative bacillus, *Salmonella enterica* serovar Typhi [1]. It is estimated that over 9 million people worldwide contracted typhoid in 2019, of whom over 110,000 died [2–7]. Typhoid is common in low- and middle-income countries [5,6]. In 2019, Malawi experienced an estimated 6,639 cases, 115 deaths, and 8,787 disability-adjusted life-years lost to typhoid [2]. A recent surveillance study in Blantyre, Malawi estimates an incidence per 100,000 person-years of 477 (95% credible interval: 372–770) [8]. With the global advent of multidrug-resistant typhoid [9,10], typhoid is becoming more challenging to treat, especially in low-income countries that often have reduced access to blood culture diagnostics or second-line therapeutics. The burden of typhoid can be lessened by improved water and sanitation and, importantly, also by vaccination [7]. In randomized, controlled vaccine trials in Nepal and in Malawi, a single dose of typhoid conjugate vaccine (TCV) had 81.6% (95%CI 58.8, 91.8) and 80.7% (64.2, 89.6) efficacy in preventing typhoid bacteraemia in children, respectively [11,12].

In April 2018, the World Health Organization recommended the introduction of TCV in typhoid-endemic countries [13]. Malawi currently plans to introduce TCV in late 2022 as a single dose at 9 months of age with a catch-up campaign for children aged 9 months to 15 years. Vaccine introduction requires investment and poses opportunity costs. Determining the cost-effectiveness of TCV is essential for appropriate investment, introduction, and evaluation. A key component of cost-effectiveness evaluation is the cost of typhoid episodes from the perspectives of the government healthcare provider and of households. Several typhoid cost of illness studies have been conducted in Asia, but there are few cost of illness data from Africa apart from a costing study of 17 typhoid cases in Tanzania [14,15]. We conducted a prospective facility-based costing cohort study to estimate the cost of illness among children and adults with laboratory-confirmed typhoid presenting to primary healthcare and referral-level government facilities in Blantyre, Malawi.

## Methods

### Study site, population, and design

Malawi is a low-income country with free primary and referral-level government-provided healthcare, including cost of drugs, laboratory tests and procedures, staff, and lodging for admitted patients. We recruited participants from three government health facilities that have established blood culture surveillance capacity: Queen Elizabeth Central Hospital (QECH), the largest hospital in Malawi and the only government hospital for Blantyre district's 1.3 million population, with inpatient bed capacity of 1200 and extensive outpatient care services [16]; Ndirande Health Centre, the largest of Blantyre's 32 health centers; and Zingwangwa Health Centre, a moderately sized urban primary health center [17].

Children and adults presenting to the three sites with reported fever ≥72 hours or objective fever ≥38.5˚C on presentation had blood drawn for bacterial culture in accordance with existing surveillance guidelines [17]. Patients with laboratory-confirmed typhoid were invited to participate in the study. Patients who usually resided outside Blantyre district, who were unable to provide written informed consent, who were admitted to another hospital for at least 24 hours before presentation to the study site, and who had a terminal illness (e.g., cancer) were not eligible to participate.

## Data collection

Electronic data capture by trained research nurses and fieldworkers occurred over a total of 3 interviews. Interviews took place at recruitment facilities (with care to ensure privacy and confidentiality, and with daily chart review until discharge for admitted patients) and at days 30 and 90 at participants' homes to gather information on post-discharge ongoing illness-related expenditures and to assess housing quality and amenities. Data validation and data cleaning were done in real time throughout the data collection period. Identifiable data were not recorded on the electronic case reporting forms. Data were stored on secure, ethics committee restricted-access-approved servers, in compliance with data management guidelines of the Malawi-Liverpool-Wellcome Trust Clinical Research Programme.

## Household costs

Direct medical, direct non-medical, and indirect costs incurred by the individual from symptom onset of the current typhoid illness were collected. Direct medical costs comprised the sum of all expenditures from symptom onset to full convalescence for all consultations, drugs, and investigations. Direct non-medical costs included transportation to and from the health facility and subsistence for the patient, parent (or family member acting as full-time hospital care-attendant), and family visitors. Indirect costs included the parent's self-reported income lost while taking care of child participants, or self-reported lost earning of individual adult participants. Information was gathered for the duration of illness, from symptom onset to full convalescence and included pre-facility healthcare seeking visits, the visit to the recruiting facility, and the post-facility period.

## Healthcare provider costs

The cost of staff managing admitted patients was calculated per bed day based on the overall total daily salary of all staff cadres serving each hospital ward to which participants were admitted. Costs included staff covering all shifts per 24-hour period, divided by beds per ward, and then multiplied by the length of a participant's hospital stay in a given ward for admitted participants. Hotel costs such as food, laundry, electricity, water, and security were based on actual hospital expenditures per bed multiplied by individual length of stay. Unit costs associated with each hotel cost were collected from the QECH hospital administrator using direct itemized costing of actual expenditures. Drug doses dispensed and their formulation were recorded on a daily basis. All drugs and medical supplies used in government health facilities are purchased from Malawi Central Medical Stores Trust with an itemized pricelist. Laboratory investigation costs were calculated by multiplying the unit cost of laboratory tests by the type and number of such investigations performed in individual participants.

## Household socioeconomic survey

Household economic surveys were conducted during follow-up home visits. Information on monthly income by source, itemized monthly expenditure, housing quality, amenities, and owned fixed assets were collected.

## Sample size and analysis

Using the number of confirmed typhoid cases in Blantyre in the previous years, we anticipated to recruit at least 200 laboratory-confirmed typhoid cases. This would have provided capacity across a range of plausible mean costs to estimate mean cost to within a margin of error of ± 10%. Data were analyzed using Stata version 17.1. Participant characteristics were

summarized with proportions and means with standard deviations as appropriate. Household and healthcare provider costs collected in Malawi Kwacha are presented in 2020 US dollars based on Reserve Bank of Malawi mid-exchange rate as of March 2020 and reported as mean with 95% confidence bounds. No *a priori* hypothesis testing was planned. Complete case analysis was done, we did not impute data for missing follow-up visits. Prior to analysis, data were examined visually for outliers, these were confirmed against clinical status (e.g. intensive care admission is more costly) and other related expenditures in each case to ensure erroneous terminal 0's did not skew data by an order of magnitude.

### Ethics

All adult participants (18 years and older) and parents/guardians of participating children provided written informed consent. Participating children aged 8 years or older provided written informed assent. Ethical approval was obtained from the University of Malawi College of Medicine Research Ethics Committee (Protocol P.03/18/2372) and by the Institutional Review Boards of Johns Hopkins University (Protocol 8895) and PATH (Protocol 1199169–3).

## Results

### Participants

Between 1st July 2019 and 20th March 2020, 109 cases of laboratory-confirmed typhoid were recruited (Table 1). Due to the COVID-19 pandemic, recruitment of new participants stopped on advice of the Sponsor and relevant Institutional Review Boards on 29th March 2020. We switched from face-to-face interviews to phone-call interviews to minimize potential coronavirus exposure to our participants and study staff. All participants were alive at day 30. Ninety-six participants (88%) were contactable at day 90, the remainder were lost to follow-up. The mean age for all study participants was 14 years (standard deviation (SD) 11.6) with 58% younger than 15 years. Females constituted 49.5% of the cohort, but among admitted adult inpatients there was a male preponderance (16/23, 69.6%). Sixty percent of participants were managed as outpatients, while for admitted inpatients, mean length of stay was 7.7 (SD 4.1) days. One adult and one child were admitted to intensive care. The adult was admitted for inotropic support following laparotomy. Another child was admitted to a high dependency unit and received oxygen support. One child had concurrent severe acute malnutrition and was managed in a specialised nutritional care unit. All patients survived, and no long-term sequelae were observed.

### Household costs

Despite mean household membership of 5.0 (SD 1.9) persons, 27 (24.8%) participants were living in houses with one sleeping room and only two participants (1.8%) were living in houses with four sleeping rooms. Mean (95% confidence bounds) monthly household income and expenditure was $107.83 (90.66, 125.01) and $63.85 (52.98, 74.73), respectively. Mean monthly household income per household member was $23.71 (19.62, 27.81) or equivalently $0.78 (0.64, 0.91) per person per day. In 27 (24.8%) households, monthly expenditure exceeded income (Table 1). Mean total household illness costs for inpatients and for outpatients respectively were $93.85 (68.87, 118.84) and $19.05 (4.38, 33.71). Mean direct medical costs for households were $2.21 (1.04, 3.38) for inpatients and $1.13 (0.48, 1.78) for outpatients. Direct non-medical costs for inpatients and outpatients respectively were $36.97 (27.41, 46.53) and $1.12 (0.54, 1.70). Self-reported indirect costs were widely variant, but were respectively $57.47 (33.52, 81.42) and $21.34 (3.26, 39.42) (Table 2).

**Table 1. Participant characteristics.**

|  |  | Inpatients | Outpatients |
|---|---|---|---|
| **Children** | | | |
| N | | 21 | 42 |
| Mean age in years (SD) | | 6.6 (4.1) | 7.0 (3.7) |
| Male (%) | | 10 (47.6) | 20 (47.6) |
| Informant: parent | | 18 (85.7) | 40 (95.2) |
| Mother deceased | | 0 | 1 |
| Maternal highest education | | | |
| | No formal education | 2 (9.5) | 6 (14.3) |
| | Primary school | 8 (38.1) | 15 (35.7) |
| | Secondary school | 11 (52.4) | 20 (47.6) |
| | Tertiary education | 0 | 1 (2.4) |
| Father deceased | | 1 | 1 |
| Paternal highest education | | | |
| | No formal education | 2 (9.5) | 5 (11.9) |
| | Primary school | 5 (23.8) | 13 (31.0) |
| | Secondary school | 13 (61.9) | 19 (45.2) |
| | Tertiary education | 1 (4.76) | 5 (11.9) |
| Mean number of persons in household (SD) | | 5.2 (1.4) | 4.9 (2.0) |
| Mean number of persons per room (SD) | | 2.6 (0.9) | 2.9 (1.3) |
| Mean days of hospital stay (SD) | | 6.1 (3.9) | NA |
| Mean total household monthly income* (SD) | | 120.46 (67.15) | 88.07 (59.39) |
| Mean total household monthly expenditure* (SD) | | 44.98 (30.79) | 73.25 (62.93) |
| **Adults** | | | |
| N | | 23 | 23 |
| Age | | 23.6 (7.2) | 28.6 (10.4) |
| Male | | 16 (69.6) | 9 (39.1) |
| Informant: self | | 18 (78.3) | 21 (91.3) |
| Mean number of persons in household (SD) | | 5.4 (2.5) | 4.6 (1.5) |
| Mean number of persons per room (SD) | | 2.5 (0.9) | 2.4 (1.1) |
| Mean days of hospital stay (SD) | | 6.5 (3.0) | NA |
| Mean total household monthly income* (SD) | | 141.83 (119.18) | 100.75 (90.18) |
| Mean total household monthly expenditure* (SD) | | 65.63 (58.02) | 64.22 (40.19) |

*2020 US dollars.

Direct medical costs accounted for 9.2%, direct non-medical for 50.8%, and indirect costs for 41.9% of total household losses. Despite available free healthcare, in 48 (44.0%) households, total costs of illness exceeded 40% of reported monthly non-food expenditure, a metric of catastrophic costs [18]. Further, in 17 (15.6%) households, illness costs exceeded total monthly income (Fig 1). Ratio of mean total illness household costs to mean total monthly household income was 1.2 (1.7) and 0.3 (0.7) for inpatients and outpatients, respectively.

## Healthcare provider

Overall mean (95% confidence bounds) cost of inpatient healthcare was $296.52 (225.79, 367.25), but was substantially higher for adult than for child inpatients (Table 2). The drivers of mean costs of inpatients for the healthcare provider in decreasing order were laboratory investigations $164.73, staffing $84.10, drugs $36.82 and hotel costs $12.83. Mean cost of

**Table 2. Mean (95% confidence bounds) household and healthcare provider costs (2020 US dollars).**

| Household costs | Inpatients | Outpatients |
|---|---|---|
| **Children** | | |
| Direct medical costs | 2.36 (0.38, 4.35) | 1.11 (0.36, 1.85) |
| Direct non-medical costs | 28.09 (19.48, 36.70) | 1.13 (0.32, 1.93) |
| Indirect costs | 47.00 (20.00, 74.00) | 28.14 (0, 58.50) |
| **Adults** | | |
| Direct medical costs | 2.07 (0.61, 3.54) | 1.17 (0, 2.47) |
| Direct non-medical costs | 45.08 (28.53, 61.62) | 1.11 (0.32, 1.91) |
| Indirect costs | 66.52 (26.63, 106.40) | 11.63 (0.69, 22.57) |
| **Healthcare provider costs** | | |
| **Children** | | |
| Hotel | 13.91 (8.39, 19.42) | 0.03 (0.02, 0.04) |
| Drug | 20.04 (6.60, 33.47) | 0.23 (0.08, 0.37) |
| Staff | 81.60 (49.68, 113.52) | 2.69 (1.54, 3.84) |
| Investigations | 118.50 (74.01, 162.99) | 36.96 (30.26, 43.66) |
| Total | 230.15 (160.93, 299.38) | 40.71 (33.24, 48.19) |
| **Adults** | | |
| Hotel | 11.80 (9.58, 14.02) | 0.02 (0.02, 0.02) |
| Drug | 52.83 (26.48, 79.19) | 0.23 (0.07, 0.38) |
| Staff | 86.37 (33.22, 139.52) | 2.02 (0.93, 3.12) |
| Investigations | 208.87 (142.24, 275.50) | 34.98 (25.83, 44.14) |
| Total | 359.87 (238.44, 481.30) | 37.77 (28.41, 47.13) |

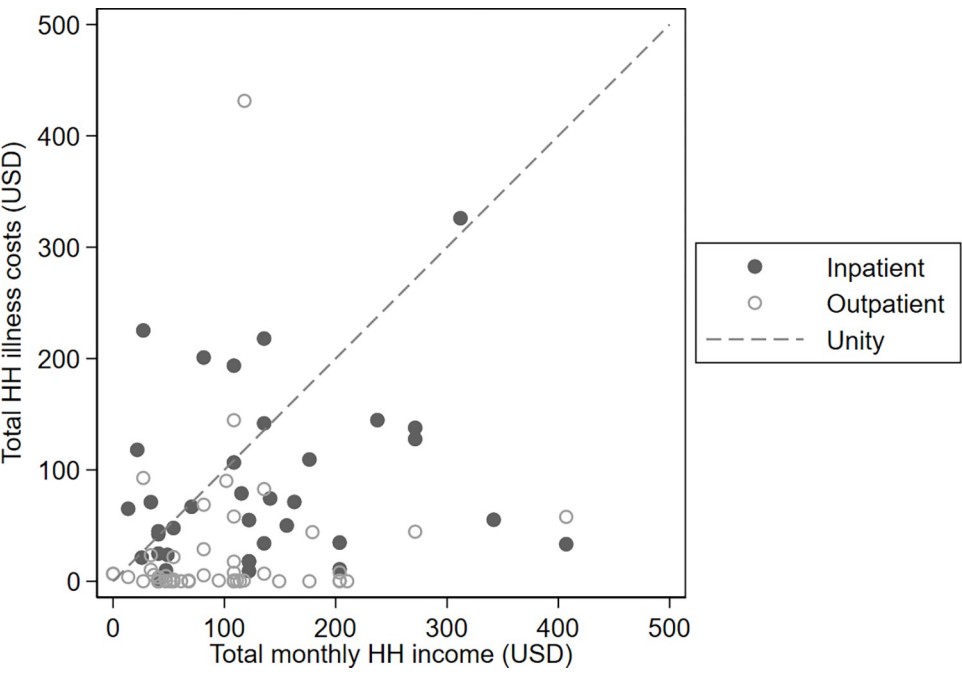

**Fig 1. Total household illness costs vs total monthly household income in 2020 US dollars.**

outpatient healthcare was $39.67 (33.96, 45.39). Mean and median costs differed because of several outliers with complicated disease, two requiring critical care admission, one with small bowel perforation, and one with obtundation and suspected meningitis.

## Sensitivity analyses

Participants reported illness-associated work time lost in addition to reporting illness-associated income loss. We recalculated loss of income by multiplying reported household income by reported time lost off work. Using this method, indirect costs were $62.71 (32.81, 92.61) for inpatients and $45.06 (15.78, 74.34) for outpatients, in both cases higher than self-reported losses. Separately, we used World Bank estimates for annual gross domestic product per capita for Malawi of $440.50 to derive daily loss and multiplied this by reported days worktime lost. Using this method, monthly income per person was $31.67 and loss of income for inpatients and outpatients was $14.97 (10.99, 18.94) and $12.18 (5.53, 18.81), respectively.

## Discussion

We have quantified the costs of typhoid for adult and child inpatients and outpatients in urban Malawi, a low-income country with high typhoid burden [19,20]. The age range of cases, with median age 15, closely resembles other reported sequential case series, indicating that our case-mix is representative [10]. Despite high uptake of Malawi's free government-provided healthcare being evidenced in the reported low direct medical costs for households, illness still resulted in substantial direct non-medical costs and income loss that were catastrophic for a large proportion of households. For outpatients, indirect costs due to loss of income were substantial. For inpatients, both direct non-medical costs and indirect costs were drivers of total cost. In a study of cholera in southern Malawi, indirect costs were also major drivers of total household costs [21] with many households consequently falling into debt.

In Malawi, household surplus savings are uncommon, and expenditure is dependent upon recent income [22,23]. Few families have the financial buffer to absorb catastrophic costs of illness. In our cohort, most participants reported reducing expenses, using savings, and incurring debt to cover typhoid illness costs. Borrowing money has economic consequences on households. Families may fall into debt, forgo basic necessities such as education for their children, or consume fewer meals a day [24]. Malawi has made major investments in making free primary healthcare accessible to the vast majority of the population, as is reflected in direct non-medical costs for outpatients. Yet for typhoid severe enough to require hospital admission, direct non-medical costs remain high. Loss of income is high even for outpatients, as was also evident in the only other typhoid costing study in East Africa from Pemba, Tanzania [15]. In Jakarta, Indonesia, the mean household cost of an outpatient was 23% of monthly income and exceeded monthly income in all hospitalized patients [14]. Our findings were similar. This economic burden occurring in the face of widely accessible free health infrastructure that provides appropriate diagnostic and therapeutic care argues for the potential value of introduction of typhoid prevention measures such as vaccination with typhoid conjugate vaccines. By preventing illness and hospitalization and their associated costs, typhoid vaccination would protect households from impoverishment and help ensure societal productivity.

From a healthcare provider perspective, the cost to the Malawian government is substantial for typhoid. On a per-episode basis, the cost of typhoid exceeds the costs of other–admittedly more common–childhood illnesses we have studied in the same context [25,26]. Hospital investigations were major contributors, in part driven by blood culture costs, and these latter investigations may not be ubiquitously available where suspected (yet unconfirmed) typhoid cases occur. Health system planners may wish to consider diagnostic costs as mitigating

potential future complication costs among those in whom disease is missed and undertreated. These findings differ from those seen in India, China, and Indonesia, where staff cost was higher [14]. This may be due to higher pay and greater staff-to-patient ratios in those contexts compared to Malawi. The highest costs were incurred for inpatients with severe disease, an outcome not uncommon in Malawi's context [27]. Although few such outlying cases may occur, they contribute to the overall cost burden and should be planned for in facility budgeting.

Although noted post hoc, we observed a higher representation of males among hospitalized inpatients (Table 1). We hazard to conjecture but by design cannot prove, that this could represent possible delay in presentation to health care centers among male breadwinners, leading to increased severity at presentation that warrants admission. This requires qualitative confirmation but is consistent with past reports of male preponderance in admission and in fatality in our setting. If correct, it serves to highlight further the complex bidirectional interplay between illness and impoverishment [28]. Care seeking is rationally delayed if perceptions of financial loss are borne out.

Our study stopped recruitment prematurely due to the COVID-19 pandemic. This means our cost estimates are less precise than we had planned, though the overall cost drivers were consistent among the participants we recruited. Despite early stopping, our recruited sample size is of similar magnitude to that of many other costing studies [14,15]. We did not value school days lost due to typhoid illness in a study population whose majority were school-going children. We relied on self-reported income and expenditure and note that income was about 75% of Malawi's reported gross national income per person. This could be because persons with foodborne illness represent a poorer sector of the population, or the discrepancy may arise from complex reporting or respondent biases or expectations among our participants. Study staff were well trained, mindful of the sensitive nature of financial questioning and issues around accuracy, and therefore probed carefully. Self-reported losses were somewhat lower than the loss we calculated based on income days lost, suggesting reporting bias if present was modest. In a cash economy such as Malawi, confirming through observed receipts or bank statement audits is all but impossible.

A study examining the cost of typhoid conjugate vaccine (TCV) introduction is currently underway in Malawi. Together with the impressive efficacy demonstrated by recent vaccine trials, our findings will help quantify TCV cost-effectiveness. When health funding is limited, planners must spend available funds wisely and choose among competing needs. Investment in prevention of infectious diseases through vaccination is not only an inherent good, but also provides other benefits beyond immediate disease prevention to families and society writ large. Such benefits include prevention of impoverishment and prevention of lost schooling and education, though these additional benefits, or indeed indirect effects through mitigation of community carriage, are not commonly incorporated into evaluation of total value of vaccines. Typhoid is uncommon in wealthy countries, even in absence of vaccination. Through harnessing economic development, vaccination may well usher in benefits well beyond disease prevention.

## Acknowledgments

We thank participants and their families. We thank our study nurses and field officers, clinic staff, and MLW support staff. We thank the Typhoid Vaccine Acceleration Consortium (TyVAC), a partnership between the Center for Vaccine Development and Global Health at the University of Maryland School of Medicine, the Oxford Vaccine Group at the University of Oxford, and PATH, an international non-profit.

## Author Contributions

**Conceptualization:** Fumbani Limani, James Meiring, Frédéric Debellut, Clint Pecenka, Melita A. Gordon, Naor Bar-Zeev.

**Data curation:** Fumbani Limani, Richard Wachepa, Naor Bar-Zeev.

**Formal analysis:** Fumbani Limani, Frédéric Debellut, Clint Pecenka, Naor Bar-Zeev.

**Funding acquisition:** Clint Pecenka, Naor Bar-Zeev.

**Investigation:** Fumbani Limani, Hlulose Chafuwa, Pratiksha Patel, Priyanka D. Patel, Naor Bar-Zeev.

**Methodology:** Christopher Smith, James Meiring, Frédéric Debellut, Melita A. Gordon, Naor Bar-Zeev.

**Project administration:** Fumbani Limani, Patrick Noah, Melita A. Gordon, Naor Bar-Zeev.

**Resources:** Christopher Smith, Frédéric Debellut, Clint Pecenka.

**Software:** Richard Wachepa.

**Supervision:** Fumbani Limani, Hlulose Chafuwa, James Meiring, Pratiksha Patel, Priyanka D. Patel, Melita A. Gordon, Naor Bar-Zeev.

**Validation:** Melita A. Gordon.

**Visualization:** Naor Bar-Zeev.

**Writing – original draft:** Fumbani Limani.

**Writing – review & editing:** Fumbani Limani, Christopher Smith, Richard Wachepa, Hlulose Chafuwa, James Meiring, Patrick Noah, Pratiksha Patel, Priyanka D. Patel, Frédéric Debellut, Clint Pecenka, Melita A. Gordon, Naor Bar-Zeev.

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
