## [Decision Letter · Decision Letter 0]

30 Aug 2022

PONE-D-22-03057Estimating the economic burden of typhoid in children and adults in Blantyre, Malawi: a costing cohort studyPLOS ONE

Dear Dr. Bar-Zeev,

Thank you for submitting your manuscript to PLOS ONE. After careful consideration, we feel that it has merit but does not fully meet PLOS ONE’s publication criteria as it currently stands. Therefore, we invite you to submit a revised version of the manuscript that addresses the points raised during the review process.

We look forward to receiving your revised manuscript.

Kind regards,

Tai-Heng Chen, M.D.

Academic Editor

PLOS ONE

Journal Requirements:

“This work was supported by a grant from PATH Seattle (https://www.path.org/) to NBZ, as part of the Typhoid Vaccine Acceleration Consortium (TyVAC). TyVAC is a partnership between the Center for Vaccine Development and Global Health at the University of Maryland School of Medicine, the Oxford Vaccine Group at the University of Oxford, and PATH, an international non-profit. TyVAC is funded by the Bill & Melinda Gates Foundation (https://www.gatesfoundation.org/) (OPP1151153). The funders had no role in study design, data collection and analysis, decision to publish, or preparation of the manuscript.”

“I have read the journal's policy and the authors of this manuscript have the following competing interests: NBZ is in receipt of research grants from Johnson & Johnson and the Serum Institute of India for work outside of Malawi entirely unrelated to this manuscript. Other authors declare that no competing interests exist.”

We note that one or more of the authors are employed by a commercial company: Johnson & Johnson and the Serum Institute of India

Reviewers' comments:

Reviewer's Responses to Questions

**Comments to the Author**

1. Is the manuscript technically sound, and do the data support the conclusions?

Reviewer #1: Yes

Reviewer #2: Partly

2. Has the statistical analysis been performed appropriately and rigorously? 

Reviewer #1: Yes

Reviewer #2: Yes

3. Have the authors made all data underlying the findings in their manuscript fully available?

Reviewer #1: Yes

Reviewer #2: Yes

4. Is the manuscript presented in an intelligible fashion and written in standard English?

Reviewer #1: Yes

Reviewer #2: Yes

5. Review Comments to the Author

Reviewer #1: Very clearly written. One small issue is the use of the word catastrophic as this is a contentious one in WHO used by health financing team as well as TB department. Can the timeframe of one month income lost be catastrophic?

Reviewer #2: The paper is written and explained in a very professional scientific style to be understood in a very simple way.

The article covers an important topic of Typhoid burden in Malawi. Many families suffer from catastrophic economic impact normally described as ‘out of pocket expenditure’ in spite of free medical care provided. This is due the direct medical and non-medical and indirect costs incurred on the typhoid illness at household and healthcare provider’s level.

Line 81-82 - “Several typhoid cost of illness studies have been conducted in Asia”, this literature review part of introduction may include the few costs related data from such studies to enlighten the present study significance and requirement for such estimates. Though discussed later in discussion parts.

Line 147- How the Sample size of 200 laboratory confirmed typhoid cases arrived at based on last years confirmed cases? Then sampling technique used to recruit participants or all reported laboratory confirmed cases included.

Line 155 – Ethics part may add how the privacy, confidentiality of data including validity ,missed, outliers data etc. were managed.

Results –

Line 164 – “Between 1st July 2019 and 20th March 2020, 109 cases of laboratory-confirmed typhoid were

165 recruited (Table 1).” However, Table 1 – N reflects 21 inpatients and 42 outpatients. The number of outpatients visits to healthcare settings is not clear.

Typhoid mortality was nil in recruited participants but how many participants develop complications of typhoid.

Discussion –

Line 300 – Since WHO has already recommended use of single dose of Typhoid Conjugate Vaccine at 9 months of age this part should highlight probable estimates how much costs /out of pocket expenditure on typhoid disease may be reduced if Typhoid Conjugate Vaccine is included in routine immunisation program of Malawi. The international cost of single dose typhoid conjugate vaccine may also be included to make audience aware.

Typhoid carrier state in community, an epidemiological dangerous aspect may also figure in discussion part which will also be prevented ,cutting further the economic burden of typhoid disease.

6. PLOS authors have the option to publish the peer review history of their article (what does this mean?). If published, this will include your full peer review and any attached files.

Reviewer #1: **Yes: **Andrew Siroka

Reviewer #2: **Yes: **Prof. Dr. Neeraj Bedi

---

## [Author Response · Author response to Decision Letter 0]

20 Sep 2022

Sept 16, 2022

Dear Editor Tai-Heng Chen and editorial team,

Re: PONE-D-22-03057

Estimating the economic burden of typhoid in children and adults in Blantyre, Malawi: a costing cohort study

Herewith our response to Journal editorial requirements and to Reviewer’s comments. We address each in turn. We would be happy to revert with any further changes that are required, and in particular seek guidance on the question of the Competing Interests as outlined in the relevant sections below.

Many thanks

Naor Bar-Zeev on behalf of all authors.

Journal Requirements:

>>> Author’s response:

We have follow revised the manuscript to meet PLOS ONE’s style requirements.

“This work was supported by a grant from PATH Seattle (https://www.path.org/) to NBZ, as part of the Typhoid Vaccine Acceleration Consortium (TyVAC). TyVAC is a partnership between the Center for Vaccine Development and Global Health at the University of Maryland School of Medicine, the Oxford Vaccine Group at the University of Oxford, and PATH, an international non-profit. TyVAC is funded by the Bill & Melinda Gates Foundation (https://www.gatesfoundation.org/) (OPP1151153). The funders had no role in study design, data collection and analysis, decision to publish, or preparation of the manuscript.”

>>> Authors’ response: 

We have provided *all* sources of funding and support. We have added the sentence “There was no additional external funding received for this study.”

“I have read the journal's policy and the authors of this manuscript have the following competing interests: NBZ is in receipt of research grants from Johnson & Johnson and the Serum Institute of India for work outside of Malawi entirely unrelated to this manuscript. Other authors declare that no competing interests exist.”

We note that one or more of the authors are employed by a commercial company: Johnson & Johnson and the Serum Institute of India

>>> Authors’ response:

This is not correct. No author is employed by these or any pharmaceutical company. NBZ is in receipt of research grants to Johns Hopkins University from Johnson & Johnson and from the Serum Institute of India, in both cases as co-investigator, for separate work occurring outside Malawi and entirely unrelated to this manuscript. We have rephrased Competing Interests section as follows: “NBZ is co-investigator on research grants to Johns Hopkins University from Serum Institute of India and from Johnson & Johnson, both outside this work.”

>>> Authors’ response:

NBZ was a co-investigator on research grants to his university from the mentioned entities. He did not and does not have any commercial affiliation with any funder. This is not a correct representation of the relationship. I would ask the editor for advice on the correct framing of this issue please.

>>> Authors’ response:

NBZ was co-investigator on research grants to his university, he was not in receipt of salary from these funders. It is incorrect to state the latter in the Funding Statement. We have not amended the Funding Statement, and seek editorial guidance on this issue given our clarification as to the nature of the relationship.

>>> Authors’ response:

As mentioned above, this was not a commercial affiliation, and it would be incorrect to state that it was. We have disclosed all the funding sources during the life of the study for full disclosure. But we would ask that the nature of the funding be correctly described.

>>> Authors’ response:

Ethics statement is in Methods section.

>>> Authors’ response:

Reference list is complete and correct.

Reviewers' comments:

Reviewer's Responses to Questions 

Comments to the Author

1. Is the manuscript technically sound, and do the data support the conclusions?

Reviewer #1: Yes

Reviewer #2: Partly

>>> Authors’ response: 

Thank you. Since no further details are given by Reviewer #2 we are uncertain how to address this concern.

2. Has the statistical analysis been performed appropriately and rigorously? 

Reviewer #1: Yes

Reviewer #2: Yes

3. Have the authors made all data underlying the findings in their manuscript fully available?

Reviewer #1: Yes

Reviewer #2: Yes

4. Is the manuscript presented in an intelligible fashion and written in standard English?

Reviewer #1: Yes

Reviewer #2: Yes

5. Review Comments to the Author

Reviewer #1:

“Very clearly written.”

>>> Authors’ response: 

Thank you.

“One small issue is the use of the word catastrophic as this is a contentious one in WHO used by health financing team as well as TB department.”

>>> Authors’ response: 

The question of definition of ‘catastrophic costs’ and the ability to distinguish poverty affects from health expenditure effects is a fecund area of ongoing research and discussion. We acknowledge this, and understand that nomenclature may be applied differently by different users. See for example the exchange between Suzanne Duryea of the Inter-American Development Bank and Ke Xu of the WHO in Health Affairs Nov/Dec 2007 (https://doi.org/10.1377/hlthaff.26.6.1789). We have used the “the more standard definition” (quoting Xu, op cit.) which has validity across settings, and we have provided the reference to our definition. The reference we used is also used by important studies in Bangladesh, in Vietnam and the Mekong Delta Subregion, and in multi-country evaluations, respectively: 

JAM Khan, S Ahmed, TG Evans. Catastrophic healthcare expenditure and poverty related to out-of-pocket payments for healthcare in Bangladesh-an estimation of financial risk protection of universal health coverage. Health Policy Plan. 2017 Oct 1;32(8):1102-1110.

S Ahmed, S Szabo, K Nilsen. Catastrophic healthcare expenditure and impoverishment in tropical deltas: evidence from the Mekong Delta region Int J Equity Health. 2018 Apr 27;17(1):53.

K Xu, DB Evans, K Kawabata, R Zeramdini, J Klavus, CJL Murray. Household catastrophic health expenditure: a multicountry analysis. Lancet . 2003 Jul 12;362(9378):111-7.

Readers of interest are welcome to consider the relative merits of various alternative definitions, to extend our academic understanding and more pointedly to optimize mitigating actions on this important issue. 

“Can the timeframe of one month income lost be catastrophic?”

>>> Authors’ response: 

We are not entirely clear as to the Reviewer’s intention here. “One month income lost” does not refer to a timeframe, but to an amount of money lost, which is at least as much as a household’s reported total monthly income under normal circumstances. Though we note that this is not the standard definition of catastrophic costs that we use, we do highlight the comparison of healthcare costs to income in Figure 1, and by scaling the line of unity we show costs greater than a month total household income. Formal definitions notwithstanding, if the reviewer is asking informally whether in our view a month income loss can be devastating to already impoverished families in Malawi, who have no savings buffer and for whom expenditure on that day’s food so often depends on daily income earned, then the answer is a resounding yes. We briefly discuss this on line 244. Certainly anecdotally we can attest to families going with less or no food, let alone school fees, shoes or spectacles, because of emergency health related costs to family members. Such repeated costs contribute to rates of anthropometric stunting in Malawi, and many other economic and health effects. We also note in the Discussion (line 254) that in Indonesia inpatient costs also exceeded monthly income.

Reviewer #2: “The paper is written and explained in a very professional scientific style to be understood in a very simple way. The article covers an important topic of Typhoid burden in Malawi. Many families suffer from catastrophic economic impact normally described as ‘out of pocket expenditure’ in spite of free medical care provided. This is due the direct medical and non-medical and indirect costs incurred on the typhoid illness at household and healthcare provider’s level.”

>>> Authors’ response: 

Thank you. Regarding the Reviewer’s phrase “catastrophic economic impact normally described as ‘out of pocket expenditure’” we would like to clarify that out of pocket expenditure is only categorized as catastrophic when it meets the formal definition. Not every out of pocket cost is catastrophic.

“Line 81-82 - “Several typhoid cost of illness studies have been conducted in Asia”, this literature review part of introduction may include the few costs related data from such studies to enlighten the present study significance and requirement for such estimates. Though discussed later in discussion parts.”

>>> Authors’ response: 

Thank you. As the reviewer notes, there are several studies in Asia. We reference Poulos C, Riewpaiboon A, Stewart JF et al since it covers five countries (China, Indonesia, India, Pakistan, and Vietnam), though there are others we have not referenced. We specifically did not report on the actual cost of illness data in these studies because studies in Asia have less relevance to African settings, the economies and the health systems are rather different, and one should generalize with caution from one setting to the other. The study from Tanzania was very small, we discuss it descriptively but are cautious of drawing any inference from its specific cost findings.

Line 147- How the Sample size of 200 laboratory confirmed typhoid cases arrived at based on last years confirmed cases? Then sampling technique used to recruit participants or all reported laboratory confirmed cases included.

>>> Authors’ response: 

This was the anticipated case burden, it was not a formal a priori sample size calculation aiming to achieve pre-specified statistical power. We aimed to recruit every case, and estimated how many such cases would occur on the basis of past disease burden observed. There was no sampling technique, since we recruited all hospital presenting laboratory confirmed cases, we did not recruit a sample of the cases but recruited all the cases.

Line 155 – Ethics part may add how the privacy, confidentiality of data including validity ,missed, outliers data etc. were managed.

>>> Authors’ response: 

Thank you for these important points. We have now addressed the Reviewer’s comments, but for the sake of flow of the manuscript we did not add to the Ethics section, but instead:

• To the section Data collection we have added: “Interviews took place at recruitment facilities (with care to ensure privacy and confidentiality…)” and also added: “Data validation and data cleaning were done in real time throughout the data collection period. Identifiable data were not recorded on the electronic case reporting forms. Data were stored on secure, ethics committee restricted-access-approved servers, in compliance with data management guidelines of the Malawi-Liverpool-Wellcome Trust Clinical Research Programme.”

• To the section on Sample Size and analysis we have added: “Complete case analysis was done, we did not impute data for missing follow-up visits. Prior to analysis, data were examined visually for outliers, these were confirmed against clinical status (e.g. intensive care admission is more costly) and other related expenditures in each case to ensure erroneous terminal 0’s did not skew data by an order of magnitude.”

Results –

Line 164 – “Between 1st July 2019 and 20th March 2020, 109 cases of laboratory-confirmed typhoid were

recruited (Table 1).” However, Table 1 – N reflects 21 inpatients and 42 outpatients. The number of outpatients visits to healthcare settings is not clear.

>>> Authors’ response: 

Table 1 reports separately children and adults as outpatients and inpatients. The Table shows clearly among children 21 inpatients and 42 outpatients, and among adults 23 inpatients and 23 outpatients, thus grand total 109 cases.

Typhoid mortality was nil in recruited participants but how many participants develop complications of typhoid.

>>> Authors’ response: 

We have amended Results Participants section and now state: “One adult and one child were admitted to intensive care. The adult was admitted for inotropic support following laparotomy for repair of small bowel perforation. Another child was admitted to a high dependency unit and received oxygen support. One child had concurrent severe acute malnutrition and was managed in a specialised nutritional care unit. All patients survived, and no long-term sequelae were observed.”

Discussion –

Line 300 – Since WHO has already recommended use of single dose of Typhoid Conjugate Vaccine at 9 months of age this part should highlight probable estimates how much costs /out of pocket expenditure on typhoid disease may be reduced if Typhoid Conjugate Vaccine is included in routine immunisation program of Malawi. The international cost of single dose typhoid conjugate vaccine may also be included to make audience aware.

>>> Authors’ response:

We agree with the Reviewer that cost of vaccine delivery and overall cost-effectiveness are very important for policy decision making. However, these are beyond the scope of this manuscript which is a cost of illness study. Separate studies are underway regarding cost of vaccine delivery in the Malawian context (as we mention in the Discussion, line 308), as well as work estimating cost-effectiveness, which depends on vaccine efficacy, cost of delivery and cost of illness. It is this latter element that is the subject matter of this manuscript.

Typhoid carrier state in community, an epidemiological dangerous aspect may also figure in discussion part which will also be prevented ,cutting further the economic burden of typhoid disease.

>>> Authors’ response:

Thank you. Again we do agree with the Reviewer’s insightful comments regarding possible vaccine impact on carriage and therefore indirect vaccine effects, which would serve to further increase overall population impact and cost-effectiveness. Though this is well beyond our manuscript for two reasons. First, we did not conduct a cross-sectional survey of pathogen carriage in stool samples from well persons in the community. This is a big undertaken, though may certainly be of value. Second, evaluating indirect effects is rarely incorporated into calculations of value of vaccines, though we wholeheartedly agree that reliable methods for incorporating indirect effects and downstream effects should be standardized and included in total economic evaluation of vaccines. The field of vaccine economics is moving in that direction, but these developments are well outside the scope of this manuscript. We have however added to the Discussion the following on line 315: “Such benefits include prevention of impoverishment and prevention of lost schooling and education, though these additional benefits, or indeed indirect effects through mitigation of community carriage, are not commonly incorporated into evaluation of total value of vaccines.”

---

## [Editor Report · Decision Letter 1]

27 Oct 2022

Estimating the economic burden of typhoid in children and adults in Blantyre, Malawi: a costing cohort study

PONE-D-22-03057R1

Dear Dr. Bar-Zeev,

We’re pleased to inform you that your manuscript has been judged scientifically suitable for publication and will be formally accepted for publication once it meets all outstanding technical requirements.

Kind regards,

Tai-Heng Chen, M.D.

Academic Editor

PLOS ONE
---

## [Editor Report · Acceptance letter]

15 Nov 2022

PONE-D-22-03057R1 

Estimating the economic burden of typhoid in children and adults in Blantyre, Malawi: a costing cohort study 

Dear Dr. Bar-Zeev:

I'm pleased to inform you that your manuscript has been deemed suitable for publication in PLOS ONE. Congratulations! Your manuscript is now with our production department. 

Kind regards, 

on behalf of

Dr. Tai-Heng Chen 

Academic Editor

PLOS ONE